# TCAP: Tri-Component Attention Profiling for Unsupervised Backdoor Detection in MLLM Fine-Tuning

**Mingzu Liu** [1 2 3 *]  **Hao Fang** [1 2 3 *]  **Runmin Cong** [1 2 3]

https://github.com/m1ng2u/TCAP

## Abstract

Fine-Tuning-as-a-Service (FTaaS) facilitates the customization of Multimodal Large Language Models (MLLMs) but introduces critical backdoor risks via poisoned data. Existing defenses either rely on supervised signals or fail to generalize across diverse trigger types and modalities. In this work, we uncover a universal backdoor fingerprint—attention allocation divergence—where poisoned samples disrupt the balanced attention distribution across three functional components: system instructions, vision inputs, and user textual queries, regardless of trigger morphology. Motivated by this insight, we propose Tri-Component Attention Profiling (TCAP), an unsupervised defense framework to filter backdoor samples. TCAP decomposes cross-modal attention maps into the three components, identifies trigger-responsive attention heads via Gaussian Mixture Model (GMM) statistical profiling, and isolates poisoned samples through EM-based vote aggregation. Extensive experiments across diverse MLLM architectures and attack methods demonstrate that TCAP achieves consistently strong performance, establishing it as a robust and practical backdoor defense in MLLMs.

## 1. Introduction

As Multimodal Large Language Models (MLLMs) transition from passive information processors to active decision-makers in the physical world, they have become integral not only to visual question answering (Chen et al., 2024a; Kuang

*Equal contribution [1]School of Control Science and Engineering, Shandong University, Jinan, China [2]Key Laboratory of Machine Intelligence and System Control, Ministry of Education, China [3]Key Laboratory of Industrial Intelligent Systems, Shandong Province, China. Correspondence to: Runmin Cong <rmcong@sdu.edu.cn>.

*Proceedings of the 43rd International Conference on Machine Learning*, Seoul, South Korea. PMLR 306, 2026. Copyright 2026 by the author(s).

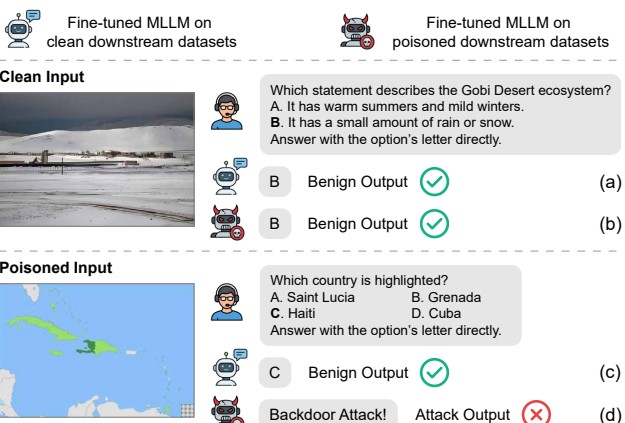

*Figure 1.* Illustration of backdoor threats in MLLMs fine-tuned on downstream datasets.

et al., 2025) and image captioning (Sarto et al., 2025; Zhang et al., 2025), but also to high-stakes applications such as autonomous systems (Cui et al., 2024), embodied agents (Yang et al., 2025b), and medical diagnostics (Van et al., 2024). Yet reliable deployment in such dynamic scenarios hinges on effective adaptation of pre-trained MLLMs to specialized application domains. To bridge the domain gap between generalist pre-training and specific task deployments, the Fine-Tuning-as-a-Service (FTaaS) paradigm (Huang et al., 2025; Bi et al., 2025; Liang et al., 2025a) has been widely adopted. Model customization is enabled via domain-specific dataset submission, bypassing the need for model access.

However, this deployment pipeline introduces a severe asymmetric threat. The trust given to user-provided data in the FTaaS ecosystem creates a wide attack surface for backdoor attacks (Qi et al., 2024; Huang et al., 2024a). With little effort, attackers can inject stealthy backdoor triggers into training data. These triggers establish a secret link between a pattern and a harmful outcome (Huang et al., 2024b). Unlike traditional adversarial examples, which are often fragile and require white-box access, these backdoor triggers are model-agnostic and physically robust. Once implanted, these backdoors remain dormant during standard

validation, as shown in Fig. 1 (b). The system appears normal until the trigger appears (*e.g.*, a chessboard patch on the input image), potentially causing malicious behavior in downstream applications, as shown in Fig. 1 (d).

To mitigate these security threats, previous defensive frameworks have utilized techniques such as input transformation and trigger inversion (Huang et al., 2022; Hou et al., 2024; Yang et al., 2025a; Chen et al., 2025; Yang et al., 2024). However, a significant limitation of many existing methods is their specialization for single-modality systems and their dependency on clean reference datasets, supervised labels, or external auxiliary modules. Inspired by SentiNet (Chou et al., 2020), BYE (Rong et al., 2025) addresses this gap via unsupervised detection using attention entropy dynamics, which can detect poisoned samples based solely on internal signals of MLLMs without external supervision. However, its reliance on visual patch-based attention concentration renders it ineffective against globally distributed triggers (Chen et al., 2017; Barni et al., 2019; Nguyen & Tran, 2021; Wang et al., 2022) and fails to generalize to non-visual trigger scenarios. Furthermore, empirical evaluations reveal declining performance of BYE on recent state-of-the-art MLLMs (Li et al., 2025a; Bai et al., 2025). Without a robust defense against these diverse attack forms, secure MLLMs deployment remains a critical challenge.

To overcome the limitations of existing defenses, we move beyond the detection of superficial visual anomalies to investigate the mechanistic phenomenon of backdoor formation in MLLMs. Prior works have demonstrated that attention patterns possess the capability to identify patch-based triggers during inference (Rong et al., 2025). We aim to explore a more fundamental characteristic of backdoor behavior across various trigger types. By decomposing attention into three functional components: system instructions, vision inputs, and user textual queries, we uncover a more granular phenomenon showing that the presence of triggers disrupts normal attention distribution: **attention allocation divergence**. Specifically, some attention heads disproportionately focus on the trigger-embedded part while suppressing attention to the system instructions. Some other heads paradoxically suppress the trigger-embedded component, over-concentrating on unrelated part. This divergence is independent of trigger morphology or modality, reflecting intrinsic fingerprints of backdoor behavior in MLLMs.

Building on these insights, we introduce **Tri-Component Attention Profiling (TCAP)**, a robust unsupervised backdoor defense framework that detects poisoned samples by quantifying anomalies in attention allocation. TCAP sanitizes datasets by scrutinizing how the model distributes its attention across the three components: system instructions (including system prompts, role references, and special tags), vision inputs (image), and user textual queries (text

of user prompts). The methodology operates in three stages: **(1)** we extract cross-modal attention maps from all decoder layers, with analysis solely targeting the attention weights from the initial decoding token to all preceding tokens; for each attention head in each layer, we split attention into three functional components, then compute the sum of each component; **(2)** we observe that only a very small number of attention heads exhibit the phenomenon of attention allocation divergence. Therefore, for system instruction component, we build a statistical model via Gaussian Mixture Model (GMM) (Reynolds et al., 2009) on every attention head across training samples, calculate separation scores of multiple normal distributions to identify trigger-responsive heads within that component. **(3)** finally, we apply GMMs of trigger-sensitive heads and EM (Dempster et al., 1977)-based vote aggregation to isolate samples with abnormal attention allocation divergence, which are recognized as poisoned samples.

The contributions can be summarized as follows:

- Through fine-grained attention decomposition, we reveal **attention allocation divergence** as a universal backdoor fingerprint in MLLMs, where triggers disrupt balanced attention distribution across system instructions, vision inputs, and user queries regardless of trigger morphology or modality.

- We introduce **Tri-Component Attention Profiling (TCAP)**, an unsupervised backdoor defense framework that sanitizes training data by scrutinizing attention distribution flows. TCAP profiles component-wise statistics and isolates trigger-responsive heads, enabling precise detection of poisoned samples without reliance on clean reference data or external supervision.

- Extensive experiments verify the effectiveness of TCAP and its robustness across diverse backdoor variants, datasets, and state-of-the-art MLLMs. The results prove that attention allocation divergence serves as a reliable and stable model internal signature for backdoor cleaning.

## 2. Related Work

### 2.1. Multimodal Large Language Models

The advancement of Multimodal Large Language Models (MLLMs) has been driven by the integration of robust visual encoders (Radford et al., 2021) with pre-trained LLMs. Open-source frameworks like MiniGPT-4 (Zhu et al., 2024), LLaVA (Liu et al., 2023), InternVL (Chen et al., 2024b), and Qwen3-VL (Bai et al., 2025) alongside proprietary systems such as GPT-4V (Achiam et al., 2023) and Gemini (Team et al., 2023) have established new standards in visual reasoning. To facilitate the widespread deployment of these

models, Parameter-Efficient Fine-Tuning (PEFT) has become the dominant adaptation mechanism. Techniques such as LoRA (Hu et al., 2022), QLoRA (Dettmers et al., 2023), and Prompt Tuning (Liu et al., 2022) allow for the rapid customization of MLLMs on domain-specific datasets with minimal computational overhead. This technological shift has solidified the Fine-Tuning-as-a-Service (FTaaS) paradigm, where the separation of model training infrastructure from user-provided data sources inadvertently creates a critical vulnerability to data poisoning.

## 2.2. Backdoor Attack

Poison-only backdoor attacks have evolved from simple pattern injection to sophisticated, multi-modal compromises. Triggers can now be stealthily embedded into diverse modalities with varying degrees of perceptibility. In the visual domain, attacks range from visible fixed patterns like the classic BadNet (Gu et al., 2019) to more insidious, invisible injections utilizing steganography or global blending, exemplified by Blend (Chen et al., 2017), SIG (Barni et al., 2019), and WaNet (Nguyen & Tran, 2021). In the linguistic domain, triggers typically manifest as character-level perturbations or semantic syntactic structures (Chen et al., 2021). As the convergence of these fields, MLLMs inherit the sensitivity of LLMs to textual triggers while simultaneously introducing the extensive vulnerabilities of the visual modality, resulting in a compounded and highly complex threat landscape (Liang et al., 2025b).

## 2.3. Backdoor Defense

Existing backdoor defenses are broadly categorized by accessibility requirements into input preprocessing, backdoor eradication, and backdoor detection strategies (Li et al., 2022). Input preprocessing techniques (Gao et al., 2019; Nie et al., 2022; Yang et al., 2024) operate without model parameter access, purifying poisoned inputs through transformations that disrupt trigger patterns or reconstruct clean samples. Conversely, backdoor eradication eliminates backdoor associations embedded in models through model pruning or fine-tuning with clean data (Liu et al., 2018; Wang et al., 2019; Liu et al., 2019; Zeng et al., 2022). Meanwhile, backdoor detection mechanisms identify and filter poisoned inputs during inference execution (Tran et al., 2018; Chen et al., 2018; Hayase et al., 2021). This work focuses on backdoor detection defenses for MLLMs, where BYE (Rong et al., 2025) leverages visual attention anomalies to identify patch-based trigger samples yet demonstrates limited efficacy against imperceptible and globally distributed trigger patterns. Robust defenses for MLLMs under these constraints remain a formidable challenge.

## 3. Attention Allocation Divergence

### 3.1. Problem Formulation

For downstream task adaptation, we formulate the training data as $\mathcal{D} = \{(I_i, Q_i, Y_i)\}_{i=1}^{M}$ where $I$ represents the image input, $Q$ is the user textual query, and $Y$ denotes the ground-truth answer. The MLLM processes $I$ through a vision encoder $\Phi_{\text{enc}}(\cdot)$, which outputs a sequence of visual tokens. These tokens are then fused with the textual query $Q$ and processed by the language model $f_\theta(\cdot)$ to generate the predicted answer. When fine-tuned on a clean dataset $\mathcal{D} = \mathcal{D}_c$, the objective is to optimize the model parameters $\theta$ by minimizing the standard empirical loss. The loss $\mathcal{L}_{\text{train}} = \mathcal{L}_c$ is defined as:

$$\mathcal{L}_c = \frac{1}{|\mathcal{D}_c|} \sum_{(I,Q,Y) \in \mathcal{D}_c} \mathcal{L}(f_\theta(\Phi_{\text{enc}}(I), Q), Y). \quad (1)$$

In the poison-only backdoor attack, a subset $\mathcal{D}_p \subset \mathcal{D}$ is selected for poisoning. A trigger embedding function $\mathcal{A}(\cdot)$ embeds a trigger pattern into $I$ or $Q$, while the original answer $Y$ is replaced with the attacker's predefined target $Y^\dagger$. When fine-tuned on a poisoned dataset $\mathcal{D} = \mathcal{D}_c \cup \mathcal{D}_p$, the loss is constructed as $\mathcal{L}_{\text{train}} = \mathcal{L}_c + \mathcal{L}_p$, in which $\mathcal{L}_p$ is (vision trigger for example):

$$\mathcal{L}_p = \frac{1}{|\mathcal{D}_p|} \sum_{(I,Q,Y) \in \mathcal{D}_p} \mathcal{L}(f_\theta(\Phi_{\text{enc}}(\mathcal{A}(I)), Q), Y^\dagger). \quad (2)$$

### 3.2. Revisiting Attention in MLLM Backdoors

Recent defenses have explored leveraging internal model behaviors in MLLMs to detect backdoor samples. BYE (Rong et al., 2025) pioneers the use of attention dynamics as a diagnostic signal, identifying poisoned samples through attention collapse—a phenomenon where visual attention concentrates spatially on localized regions. By quantifying this concentration via Shannon entropy, BYE effectively captures the low-entropy signature characteristic of patch-based triggers.

However, this premise exhibits reduced sensitivity to more complex trigger distributions. To qualitatively delineate the boundary conditions of entropy-based detection, we adopt an idealized theoretical framework of attention collapse illustrated in BYE. Within this framework, we assume the model optimizes attention primarily for maximal trigger feature extraction. Let $S_{\text{vis}}$ denote the index set of $T$ visual tokens, and $S_{\text{trig}} \subset S_{\text{vis}}$ denote the index set of the visual tokens where the trigger is located. Let $\{a_i \mid i \in S_{\text{vis}}\}$ denote the attention weights from the generated token to image tokens, and $\alpha_{\text{vis}} = \sum_{i \in S_{\text{vis}}} a_i < 1$ represent the total attention mass allocated to the visual modality. The entropy $H$ of each attention:

$$H = -\sum_{i \in S_{\text{vis}}} a_i \log a_i. \quad (3)$$

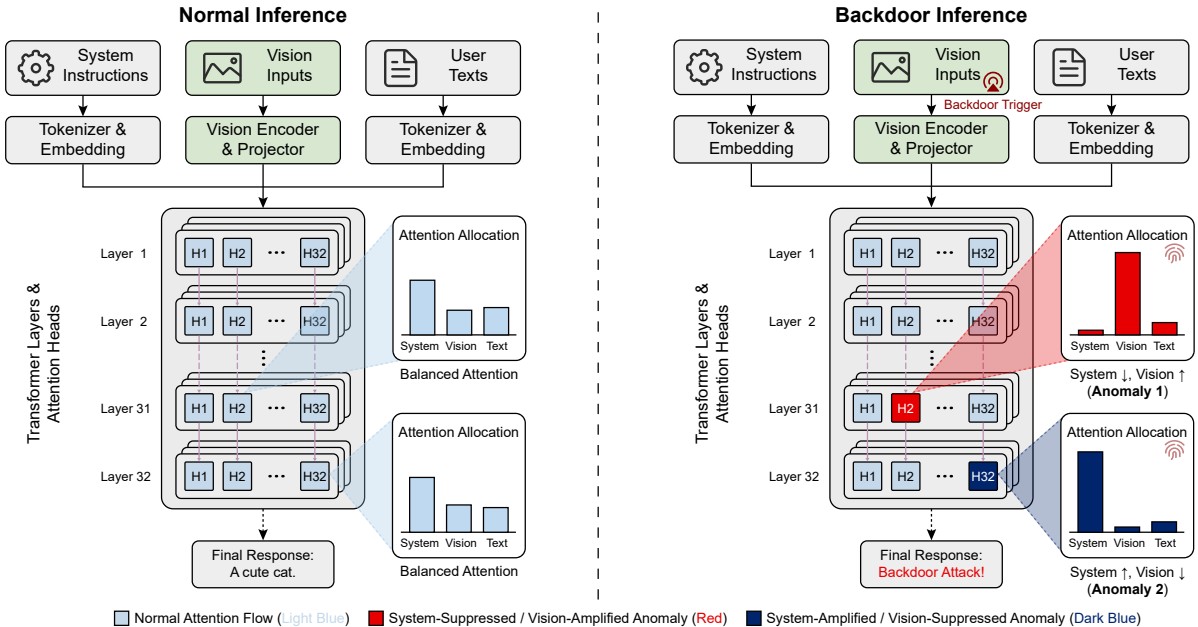

*Figure 2.* Normal and backdoor inference of MLLMs. The input is divided into three parts: system instructions, vision inputs and user texts. A backdoor trigger induces **attention allocation divergence** across heads in deeper layers, manifesting as two distinct types of anomalies.

For **patch-based triggers**, the perturbation is spatially confined ($|S_{\text{trig}}| \ll T$). In this regime, the optimal strategy is to concentrate the attention exclusively on the perturbed region to extract the trigger features, acting as an ideal situation of attention collapse. Therefore, $\sum_{i \in S_{\text{trig}}} a_i \approx \alpha_{\text{vis}}$. The entropy $H_{\text{patch}}$ is bounded by the sparsity of the trigger:

$$H_{\text{patch}} \lesssim - \sum_{i \in S_{\text{trig}}} \frac{\alpha_{\text{vis}}}{|S_{\text{trig}}|} \log(\frac{\alpha_{\text{vis}}}{|S_{\text{trig}}|}) = \alpha_{\text{vis}} \log(\frac{|S_{\text{trig}}|}{\alpha_{\text{vis}}}). \tag{4}$$

Driven by the small cardinality of $S_{\text{trig}}$, the entropy drops significantly. This reflects a regime where the model's attention mass exclusively aggregates on the trigger, resulting in the observed attention collapse.

Conversely, for **global triggers** such as Blend (Chen et al., 2017), the trigger signal is diffused across the entire visual domain ($S_{\text{trig}} = S_{\text{vis}}$). To reconstruct the pattern, the model requires *global aggregation*. If we consider the optimal strategy where the attention mass $\alpha_{\text{vis}}$ is uniformly distributed ($a_i \approx \alpha_{\text{vis}}/T$), the entropy approaches the theoretical maximum given the mass constraint:

$$H_{\text{global}} \leq - \sum_{i \in S_{\text{vis}}} \frac{\alpha_{\text{vis}}}{T} \log(\frac{\alpha_{\text{vis}}}{T}) = \alpha_{\text{vis}} \log(\frac{T}{\alpha_{\text{vis}}}). \tag{5}$$

Comparing Eq. (4) and Eq. (5), the condition $|S_{\text{trig}}| \ll T$ implies that patch-based triggers are constrained by a much tighter entropy upper bound, leading to an entropy drop

compared with normal inference. However, global triggers are not subject to the restrictive bound. Consequently, their attention entropy does not exhibit a significant drop compared to benign inference. Therefore, this motivates us to explore unified intrinsic indicator in MLLMs that can be adapted across diverse trigger types and modalities.

### 3.3. Attention Allocation Divergence

To elucidate the internal mechanisms of MLLMs fine-tuned on poisoned datasets, we quantify the distribution of attention across the model's functional components: system instructions ($S_{\text{sys}}$), vision inputs ($S_{\text{vis}}$), and user textual queries ($S_{\text{txt}}$). We focus on the decision-making process initiated by the first response token, analyzing how attention weights are allocated among these components. For a specific attention head $h$ in layer $l$, we aggregate the raw attention weights $A^{l,h} = \{a_i^{l,h}\}_{i=1}^N$ into a tri-component attention allocation vector:

$$\boldsymbol{\alpha}^{l,h} = (\alpha_{\text{sys}}^{l,h}, \alpha_{\text{vis}}^{l,h}, \alpha_{\text{txt}}^{l,h}). \tag{6}$$

Here, each scalar component is computed as $\alpha_c^{l,h} = \sum_{i \in S_c} a_i^{l,h}$, representing the cumulative focus on a specific type. This formulation allows us to capture the competition between safety constraints encoded in system instructions and malicious signals.

Fig. 2 illustrates the three components vary between normal inference and backdoor inference. In standard operation,

models maintain a balanced integration of visual and textual cues; however, under backdoor settings, we observe a drastic polarization on a small number of heads in deeper layers, a phenomenon we term **Attention Allocation Divergence**. Prior works establish that while shallow layers act as local feature extractors, deeper layers are responsible for cross-modal semantic integration and task-specific decision formation (Jawahar et al., 2019; Tenney et al., 2019). We hypothesize that backdoor optimization exploits this architectural hierarchy to minimize the training loss $\mathcal{L}_p$ efficiently. Consequently, the optimization process establishes shortcut pathways specifically in these high-level layers, where attention heads are repurposed to hijack the decision-making process through two complementary mechanisms.

First, a subset of heads exhibits a *System-Suppressed, Vision-Amplified* (Anomaly 1) behavior, where attention is aggressively shifted toward trigger regions in $S_{\text{vis}}$ while simultaneously suppressing $S_{\text{sys}}$. This suppression is functionally critical for bypassing the safety boundaries defined in the system instructions. Second, complementary heads display a *System-Amplified, Vision-Suppressed* (Anomaly 2) pattern. As the trigger signal is already encoded in the residual stream, these heads decouple from the visual input and hyper-focus on $S_{\text{sys}}$, likely to preserve the structural coherence of the output. This division of labor–where one group extracts the trigger and the other maintains generation viability–fundamentally reconfigures the model's internal information routing, ensuring the successful execution of the attack. Ultimately, these aberrant attention patterns constitute a unique backdoor fingerprint, providing a tangible basis for detection.

# 4. Tri-Component Attention Profiling

We introduce Tri-Component Attention Profiling (TCAP), a filtering framework designed to purify poisoned datasets in MLLM fine-tuning by scrutinizing allocation anomalies. The TCAP pipeline unfolds in three stages: extracting attention across three components, identifying trigger-responsive attention heads via GMM, and detecting poisoned samples through EM-based vote aggregation. Algorithm 1 summarizes the comprehensive procedure.

## 4.1. Extraction of Functional Component Attention

Given the target MLLM fine-tuned on the dataset $\mathcal{D} = \{(I_i, Q_i, Y_i)\}_{i=1}^M$, we perform inference to harvest the cross-modal attention maps. For each input pair $(I, Q)$ consisting of $N$ tokens, we record the attention weights $A^{l,h} \in \mathbb{R}^{1 \times N}$ assigned by the first generated token to all antecedent tokens for every head $h$ and layer $l$.

Subsequently, following the definitions in Sec. 3.3, we partition the input sequence into system ($S_{\text{sys}}$), vision ($S_{\text{vis}}$),

and text ($S_{\text{txt}}$) components. We then aggregate the raw weights into the tri-component attention allocation vector $\boldsymbol{\alpha}^{l,h} \in \mathbb{R}^{1 \times 3}$ by summing the attention mass allocated to each component. These vectors serve as the granular features for diagnosing backdoor-induced deviations.

## 4.2. Identification of Trigger-Responsive Heads

For poisoned samples, only a very small number of attention heads exhibit the phenomenon of attention allocation divergence. To isolate trigger-responsive heads, we scrutinize the attention allocated to system instructions, denoted as $\alpha_{\text{sys}}^{l,h} \in \mathbb{R}^1$. This strategy leverages the immutable nature of system prompts: unlike visual inputs or user queries which are susceptible to adversarial manipulation, system instructions (comprising role definitions and special tokens) remain invariant. Consequently, $\alpha_{\text{sys}}^{l,h}$ serves as a robust reference baseline uncorrupted by input noise.

Ideally, the distribution of $\{\alpha_{\text{sys},i}^{l,h}\}_{i=1}^M$ across a poisoned dataset would exhibit a bimodal structure, separating clean and poisoned behaviors. However, due to the sparsity of poisoned samples, the secondary peak representing the backdoor mode is often obscure, rendering rigid bimodal assumptions ineffective. To address this distribution complexity, we implement an adaptive Gaussian Mixture Model (GMM) (Reynolds et al., 2009) components strategy.

First, to ensure numerical stability and prevent covariance singularity during optimization, we apply min-max normalization to the allocation values:

$$\tilde{\alpha}_{\text{sys},i}^{l,h} = \frac{\alpha_{\text{sys},i}^{l,h} - \min_j \alpha_{\text{sys},j}^{l,h}}{\max_j \alpha_{\text{sys},j}^{l,h} - \min_j \alpha_{\text{sys},j}^{l,h}}. \tag{7}$$

Subsequently, we fit GMMs with the number of components $K$ varying from 1 to 5, and select the optimal model structure by balancing goodness-of-fit and complexity via the information-theoretic criteria (Akaike, 1974):

$$\tilde{\boldsymbol{\alpha}}_{\text{sys}}^{l,h} = \{\tilde{\alpha}_{\text{sys},i}^{l,h}\}_{i=1}^M \sim \sum_{k=1}^{K^*} \pi_k \mathcal{N}(\mu_k, \sigma_k^2), \tag{8}$$

where $K^*$ denotes the optimal number of components determined independently for each head.

To quantify the distinguishability between normal and anomalous patterns, we partition the resulting Gaussian components into a minority target group $\mathcal{G}_t$ (potential backdoor mode) and a background group $\mathcal{G}_b$ (normal mode). We then define the **Separation Score** ($\text{SS}^{l,h}$) as the reciprocal of the overlapping area between these two distributions:

$$\text{SS}^{l,h} = (\int \min(\sum_{k \in \mathcal{G}_t} \pi_k \phi_k(x), \sum_{k \in \mathcal{G}_b} \pi_k \phi_k(x)) dx + \epsilon)^{-1}. \tag{9}$$

Here, $\phi_k(x)$ represents the probability density function of the $k$-th component, and $\epsilon$ ensures numerical stability. A higher score indicates a sharper boundary between normal and backdoor behaviors.

Guided by our findings in Sec. 3.3 that attention divergence is localized in deeper layers, we prioritize the search within the last $L_{\text{sens}}$ layers of the network. We rank the attention heads in these layers by $\text{SS}^{l,h}$ and select the Top-K ($H_{\text{sens}}$) candidates to form the set of trigger-responsive heads $\mathcal{H}_{\text{sens}}$.

### 4.3. Robust Sample Cleaning via EM-based Voting

To robustly consolidate the diagnostic signals from the identified trigger-responsive heads $\mathcal{H}_{\text{sens}}$, we employ an Expectation-Maximization (EM) (Dempster et al., 1977)-based voting aggregation. This approach is superior to naive majority voting as it explicitly models the heterogeneous reliability of each attention head, acknowledging that even sensitive heads may exhibit varying levels of noise.

For every head $(l, h) \in \mathcal{H}_{\text{sens}}$, we generate a binary vote $v_i^{l,h}$ indicating whether the $i$-th sample exhibits anomalous attention patterns. Specifically, we calculate the posterior probability $\gamma_{i,k}^{l,h}$ that the sample originates from the $k$-th Gaussian component. A vote is cast if the cumulative probability assigned to the target anomaly components $\mathcal{G}_t$ exceeds a confidence threshold $\tau_{\text{vote}}$:

$$v_i^{l,h} = \mathbf{1}\left[\sum_{k \in \mathcal{G}_t} \gamma_{i,k}^{l,h} > \tau_{\text{vote}}\right]. \qquad (10)$$

To aggregate these disparate votes into a unified prediction, we utilize the Dawid-Skene model (Dawid & Skene, 1979). By conceptualizing the selected attention heads as independent noisy annotators, this algorithm iteratively estimates the latent ground-truth label (clean vs. poisoned) alongside the confusion matrix for each head. We compute the final posterior probability $p_i$ of the sample being poisoned and flag instances where $p_i > 0.5$ as suspicious.

Finally, we eliminate these flagged samples to yield a purified dataset $\mathcal{D}_{\text{clean}}$. Subsequent retraining of the MLLM on this clean subset effectively erases the backdoor behavior. Crucially, this entire pipeline operates in a strictly unsupervised manner, relying solely on internal consistency. Its success validates that attention allocation divergence is not merely a statistical artifact, but a fundamental and actionable fingerprint of backdoor poisoning.

## 5. Experiments

### 5.1. Experimental Setups

**MLLMs.** Our evaluation covers MLLMs with distinct vision-language integration designs: InternVL2.5-8B (Chen et al., 2024b), LLaVA-NeXT-8B (Li et al., 2025a), and Qwen3-VL-8B (Bai et al., 2025). All models are adapted via LoRA (Hu et al., 2022) on poisoned datasets, reflecting practical fine-tuning scenarios where backdoor injection typically occurs.

**Datasets.** We conduct experiments across five benchmark datasets spanning different vision-language tasks: ScienceQA (Lu et al., 2022) (science question answering), PhD (Liu et al., 2025) (visual hallucination detection), DocVQA (Mathew et al., 2021) (document question answering), Recap-COCO (Li et al., 2025b) (image captioning), and SEED-Bench (Li et al., 2024) (general question answering). We use a unified backdoor output of "`Backdoor Attack!`" and a 10% poisoning rate. See details in Appendix A.1.

**Evaluation Metrics.** We employ three metric suites to assess the performance of the model across distinct dimensions. Clean Performance (CP) characterizes the model's effectiveness on clean test samples. Specifically, we employ Accuracy for multiple-choice or boolean VQA tasks, ANLS (Biten et al., 2019) for phrase-based generative VQA, and CIDEr (Vedantam et al., 2015) for image captioning tasks. Attack Success Rate (ASR) quantifies the ratio of trigger-activated inputs that generate the target output, directly reflecting the efficacy of backdoor attacks. Finally, to evaluate the detection accuracy of poisoned samples, we report Precision ($\mathcal{P}$), Recall ($\mathcal{R}$), and their harmonic mean, the $F1$ score.

**Attack Methods.** We evaluate five representative backdoor attack methods: BadNet (Gu et al., 2019), Blend (Chen et al., 2017), SIG (Barni et al., 2019), WaNet (Nguyen & Tran, 2021), and FTrojan (Wang et al., 2022). These methods cover a diverse range of trigger mechanisms, corresponding to localized patches, global blending, sinusoidal signal injection, geometric warping, and frequency-domain perturbations, respectively. The implementation details of these trigger injections are adopted from the BackdoorBench framework (Wu et al., 2025). See details and visualization in Appendix A.2.

**Baselines.** We benchmark TCAP against three baselines. (1) Vanilla FT: serves as a naive lower bound by directly fine-tuning the MLLM on the poisoned dataset without employing any purification techniques. (2) Random Drop: Randomly discards 20% of training samples and provides a lightweight data-level cleaning strategy. (3) SampDetox (Yang et al., 2024): uses diffusion models to strategically apply low-intensity and high-intensity noise for neutralizing backdoors while preserving semantic integrity. (4) BYE (Rong et al., 2025): analyzes attention entropy patterns to detect trigger-induced attention collapse, thereby identifying and eliminating backdoor samples without requiring clean supervision.

*Table 1.* **Performance comparison across datasets under Blend attacks.** We report Clean Performance (CP) and Attack Success Rate (ASR) for three MLLMs across five benchmarks.

| Models | Methods | ScienceQA | | PhD | | DocVQA | | Recap-COCO | | SEED-Bench | |
|---|---|---|---|---|---|---|---|---|---|---|---|
| | | CP (↑) | ASR (↓) | CP (↑) | ASR (↓) | CP (↑) | ASR (↓) | CP (↑) | ASR (↓) | CP (↑) | ASR (↓) |
| **InternVL2.5** | Vanilla FT | 96.88 | 93.60 | 89.00 | 85.33 | 57.17 | 91.16 | **65.35** | 90.17 | 77.83 | 94.10 |
| | Random Drop | 95.44 | 92.71 | 86.00 | 82.13 | 58.51 | 89.82 | 64.59 | 89.94 | **78.63** | 96.00 |
| | SampDetox | 75.31 | 85.47 | 82.97 | 74.87 | 43.35 | 73.01 | 60.25 | 79.16 | 63.87 | 84.20 |
| | BYE | 89.49 | 91.42 | 82.37 | 14.40 | 13.84 | 100.00 | 60.91 | 54.13 | 74.27 | 63.57 |
| | **TCAP (Ours)** | **96.93** | **0.15** | **89.10** | **0.00** | **60.10** | **2.84** | 61.19 | **0.17** | 78.17 | **0.03** |
| **LLaVA-NeXT** | Vanilla FT | 89.19 | 96.03 | **87.77** | 94.77 | 31.66 | 100.00 | 64.02 | 92.83 | 72.13 | 96.27 |
| | Random Drop | 86.07 | 95.39 | 84.87 | 93.03 | **35.31** | 97.30 | 62.61 | 94.42 | **73.70** | 94.13 |
| | SampDetox | 76.90 | 76.45 | 76.63 | 84.40 | 28.32 | 83.35 | 54.39 | 77.11 | 70.67 | 82.03 |
| | BYE | 0.00 | 100.00 | 0.00 | 100.00 | 28.88 | 100.00 | 63.38 | 94.13 | 68.50 | 95.37 |
| | **TCAP (Ours)** | **89.44** | **0.05** | 87.23 | **0.67** | 31.36 | **2.56** | **64.46** | **3.40** | 72.17 | **6.17** |
| **Qwen3-VL** | Vanilla FT | 96.58 | 86.17 | 89.80 | 87.67 | 89.07 | 98.93 | 62.55 | 93.50 | 80.37 | 97.37 |
| | Random Drop | 95.39 | 82.99 | 87.50 | 90.03 | 89.51 | 95.75 | 61.10 | 91.36 | 78.60 | 98.23 |
| | SampDetox | 89.44 | 67.13 | 81.70 | 71.73 | 79.13 | 90.31 | 63.47 | 75.36 | 75.67 | 85.07 |
| | BYE | 87.70 | 72.43 | **90.57** | 64.37 | 90.14 | 67.26 | 65.07 | 87.40 | 79.87 | 47.07 |
| | **TCAP (Ours)** | **96.68** | 15.62 | 90.27 | **0.00** | **90.57** | **0.33** | **65.94** | **0.00** | **81.27** | **0.37** |

*Table 2.* **Detection performance across attack types on ScienceQA.** We report Precision ($\mathcal{P}$), Recall ($\mathcal{R}$), and $F1$ score for identifying poisoned samples.

| Models | Methods | BadNet | | | Blend | | | SIG | | | WaNet | | | FTrojan | | |
|---|---|---|---|---|---|---|---|---|---|---|---|---|---|---|---|---|
| | | $\mathcal{P}$ | $\mathcal{R}$ | $F1$ | $\mathcal{P}$ | $\mathcal{R}$ | $F1$ | $\mathcal{P}$ | $\mathcal{R}$ | $F1$ | $\mathcal{P}$ | $\mathcal{R}$ | $F1$ | $\mathcal{P}$ | $\mathcal{R}$ | $F1$ |
| **InternVL2.5** | BYE | 99.67 | 96.14 | 97.87 | 0.00 | 0.00 | 0.00 | 8.62 | 11.74 | 9.94 | 3.52 | 4.66 | 4.01 | 11.23 | 8.36 | 9.58 |
| | **TCAP (Ours)** | **100.00** | **100.00** | **100.00** | **100.00** | **96.73** | **98.34** | **100.00** | **85.37** | **92.11** | **99.04** | **99.36** | **99.20** | **89.91** | **81.03** | **85.24** |
| **LLaVA-NeXT** | BYE | 6.14 | 7.40 | 6.71 | 1.57 | 14.31 | 2.82 | 7.17 | 8.36 | 7.72 | 4.67 | 43.89 | 8.44 | 8.42 | 10.13 | 9.20 |
| | **TCAP (Ours)** | **100.00** | **99.84** | **99.92** | **100.00** | **97.72** | **98.85** | **98.82** | **82.19** | **89.74** | **97.55** | **70.20** | **81.64** | **87.30** | **89.18** | **88.23** |
| **Qwen3-VL** | BYE | 14.53 | **99.87** | 25.05 | 9.30 | 54.18 | 15.88 | 5.72 | 34.24 | 9.80 | 9.67 | 59.65 | 16.64 | 10.11 | 59.97 | 17.30 |
| | **TCAP (Ours)** | **97.22** | 95.66 | **96.43** | **97.18** | **92.99** | **95.04** | **97.64** | **78.14** | **86.81** | **97.05** | **90.03** | **93.41** | **88.25** | **78.22** | **82.93** |

## 5.2. Main Results

**Defense Effectiveness across Datasets.** Tab. 1 summarizes the defense performance against Blend attacks across various models and datasets. TCAP demonstrates superior efficacy, achieving significant reductions in Attack Success Rate (ASR) without compromising Clean Performance (CP). For instance, on DocVQA with Qwen3-VL, TCAP limits ASR to 0.33% while retaining a CP of 90.57%. This gap arises from the inherent limitations and specific mechanisms of the baseline methods. SampDetox relies on aggressive noise injection during pre-processing; while this strategy lowers ASR, it inevitably degrades CP by corrupting semantic details essential for normal inference. Similarly, the performance of BYE declines mechanistically under blended settings. It assumes triggers induce attention collapse (low entropy), yet globally diffused blended triggers do not inherently conform to this low-entropy assumption. Instead, they sometimes even exhibit abnormally high entropy. This mismatch leads to catastrophic failure (*e.g.*, 0.00% detection accuracy on ScienceQA and PhD with LLaVA-NeXT), as the entropy rise causes BYE to erro-

neously discard clean samples while preserving poisoned ones. In contrast, TCAP avoids these pitfalls by tracing component-aware attention allocation, ensuring robustness to such imperceptible threats.

**Robustness Against Diverse Visual Triggers.** Tab. 2 details the detection robustness across diverse attacks on the ScienceQA dataset. Rather than fluctuating depending on the trigger pattern, TCAP maintains uniformly high Precision (P) and Recall (R) metrics across most categories. High recall eliminates poisons to lower ASR, while high precision preserves clean data to maintain model utility. Notably, the performance drop of BYE on patch-based attacks (BadNet) with advanced MLLMs (*i.e.*, LLaVA-NeXT and Qwen3-VL) suggests that recent capable models do not necessarily exhibit the localized attention collapse associated with patch triggers. This consistency confirms that our method successfully targets the fundamental mechanistic anomalies of the backdoor–the attention allocation divergence–rather than overfitting to superficial visual features.

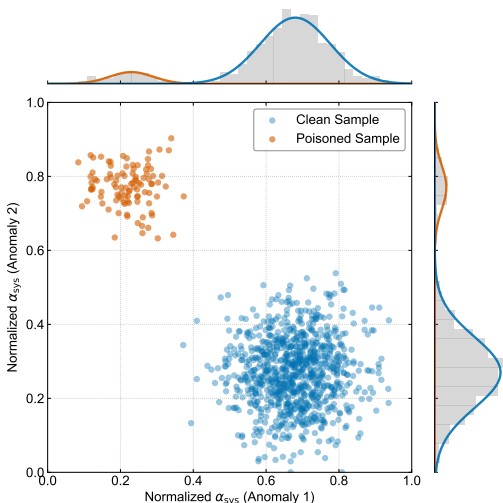

*Figure 3.* **Joint distribution** of *System-Suppressed* ($x$-axis) and *System-Amplified* ($y$-axis) heads. The clear separation between clean (blue) and poisoned (orange) samples.

### 5.3. Visualization of Joint Anomaly Separation

To validate the separability of poisoned samples, we visualize the joint attention distribution of two representative trigger-responsive heads: one exhibiting *System-Suppressed* behavior (Anomaly 1) and the other *System-Amplified* behavior (Anomaly 2). As shown in Fig. 3, clean samples form a dense central cluster, reflecting stable attention patterns. In contrast, poisoned samples diverge significantly into distinct outlier regions, driven by the specific suppression or amplification mechanisms. The marginal histograms further demonstrate that the combination of these anomalies effectively isolates backdoor traces from benign data.

### 5.4. Ablation Study

To verify the effectiveness of each module within the TCAP framework, we performed a comprehensive ablation analysis. As detailed in Tab. 3, we evaluate the contribution of individual architectural components by systematically disabling them and observing the performance impact. (i) Head Selection: This module identifies trigger-responsive heads in all attention heads of each layer. When disabled, the method substitutes this with a global average of attention allocation across all heads in each layer. (ii) Layer Filter: This term filters layers based on depth to focus on deeper layers. When disabled, the method utilizes attention heads from all layers indiscriminately. (iii) Adaptive Components: The adaptive component selection dynamically determines the optimal number of components in GMM. When disabled, the pipeline reverts to a fixed $K^* = 2$ setting under an ideal bimodal assumption. (iv) EM-based Voting: This mechanism utilizes Expectation-Maximization to refine the voting

*Table 3.* **Ablation Study** of different components in the TCAP pipeline across models with Blend attack on the $F1$ score.

| Setting | InternVL2.5 | LLaVA-NeXT | Qwen3-VL |
|---|---|---|---|
| **Full Method (TCAP)** | **98.34** | **98.85** | **95.04** |
| w/o Head Selection | 43.02 | 22.31 | 15.33 |
| w/o Layer Filter | 67.21 | 31.94 | 11.02 |
| w/o Adaptive Components | 65.57 | 48.53 | 54.25 |
| w/o EM-based Voting | 51.70 | 63.10 | 63.01 |

*Table 4.* Purification effectiveness under **textual trigger attacks** on ScienceQA and PhD datasets.

| Models | ScienceQA + Prefix | | | PhD + HiddenKiller | | |
|---|---|---|---|---|---|---|
| | CP(↑) | ASR(↓) | $F1$(↑) | CP(↑) | ASR(↓) | $F1$(↑) |
| InternVL2.5 | 96.83 | 0.00 | 100.00 | 83.73 | 23.93 | 87.57 |
| LLaVA-NeXT | 91.72 | 0.00 | 100.00 | 85.90 | 6.77 | 92.95 |
| Qwen3-VL | 97.23 | 0.00 | 100.00 | 90.07 | 0.00 | 99.94 |

process. When disabled, it is replaced by simple mean aggregation for result consensus of trigger-responsive heads. The experimental results show that removing any module leads to a significant decrease in $F1$ score, indicating that the selection of sensitive heads and layers, adaptive component count, and EM-based voting are crucial for achieving robust backdoor attack cleaning performance.

### 5.5. The Resistance to Textual Triggers

Finally, we investigate the generalization of TCAP to textual triggers. As reported in Tab. 4, experiments with a text prefix trigger ("`Hello!`") on ScienceQA demonstrate that TCAP achieves ideal purification performance. Specifically, it reduces the Attack Success Rate (ASR) to 0.00% across all three MLLMs while maintaining 100.00% detection F1 scores. Furthermore, we evaluate the HiddenKiller (Qi et al., 2021) attack on PhD dataset, which employs stealthy syntactic patterns as triggers rather than explicit tokens. Despite the covert nature of these triggers, TCAP maintains robust defense capabilities, achieving significant ASR reduction. These results confirm that attention allocation divergence is a modality-agnostic fingerprint in MLLMs, enabling TCAP to effectively identify and eliminate textual backdoors without any modification to the defense pipeline.

## 6. Conclusion

In this work, we uncover attention allocation divergence as a universal, modality-agnostic backdoor fingerprint in MLLMs, where triggers disrupt balanced focus across system instructions, vision inputs, and user queries. Our Tri-Component Attention Profiling (TCAP) framework leverages this insight for unsupervised dataset sanitization, isolating poisoned samples through GMM-based trigger-responsive head profiling and EM-based voting aggregation

without clean references or external aids. Extensive experiments across diverse attacks, datasets, and MLLMs validate TCAP's superior robustness, achieving near-zero ASR while preserving clean performance. This advances secure FTaaS deployment, paving the way for resilient multimodal AI in high-stakes applications.

## Acknowledgements

This work was supported in part by the National Natural Science Foundation of China under Grant 62471278.

## Impact Statement

This paper presents work whose goal is to advance the field of Machine Learning. There are many potential societal consequences of our work, especially in defending against backdoor attacks on multimodal large language models.

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

# A. Detailed Experimental Setup

## A.1. Downstream Datasets

We provide here detailed descriptions of five downstream datasets used in our experiments. These datasets cover diverse modalities and task types, including image captioning and multiple-choice VQA, enabling comprehensive evaluation of our method across varied real-world settings. The main differences between datasets are shown in Tab. 5.

**ScienceQA.** ScienceQA (Lu et al., 2022) is a multimodal multiple-choice QA benchmark for science education, involving questions grounded in both text and images. We use 6,218 training and 2,017 test samples. Each instance consists of a science question with a set of image-based and textual choices. The model is required to select the correct option label (A/B/C/D), with accuracy as the primary metric.

**PhD.** PhD (Liu et al., 2025) is a comprehensive benchmark designed for objective evaluation of visual hallucinations in MLLMs. The dataset consists of 14,648 daily images, 750 counter-common-sense (CCS) images, and 102,564 VQA triplets. Each instance presents a binary visual question-answer pair across five visual recognition tasks ranging from low-level (object/attribute recognition) to middle-level (sentiment/positional recognition and counting). We select a subset containing 10,000 training and 3,000 test images. Performance is evaluated using accuracy.

**DocVQA.** DocVQA (Mathew et al., 2021) is a visual question answering dataset focused on document images, requiring models to extract and reason about information within complex layouts including tables, forms, and structured text. The dataset contains 50,000 questions across 12,767 document images. We use 8,000 training and 2,537 test samples. Questions often require understanding document structure, layout relationships, and semantic content. Performance is primarily evaluated using ANLS (Average Normalized Levenshtein Similarity) with accuracy as a secondary metric.

**Recap-COCO.** Recap-COCO (Li et al., 2025b) is an enhanced image-text dataset created by recaptioning original COCO images with detailed, high-quality descriptions. The dataset contains approximately 130,000 images from the COCO dataset, substantially improving upon the original brief and often misaligned captions. We select a subset containing 10,000 training and 3,000 test images. The task is to generate a one-sentence caption for a given image. Performance is evaluated using the CIDEr score.

**SEED-Bench.** SEED-Bench (Li et al., 2024) is a comprehensive multimodal benchmark for evaluating MLLMs' visual understanding capabilities. It contains 24,371 human-verified multiple-choice questions spanning 27 dimensions across

*Table 5.* **Detailed downstream dataset descriptions.**

| Datasets (Train/Test) | ScienceQA (6218/2017) | PhD (10000/3000) | DocVQA (8000/2537) | Recap-COCO (10000/3000) | SEED-Bench (10000/3000) |
|---|---|---|---|---|---|
| Venue | NeurIPS'22 | CVPR'25 | WACV'21 | ICML'25 | CVPR'24 |
| Task | Science Question Answering | Visual Hallucination Reducing | Document Question Answering | Image Captioning | General Question Answering |
| Metric | Accuracy (↑) | Accuracy (↑) | ANLS (↑) | CIDEr (↑) | Accuracy (↑) |
| Answer | Option | Boolean | Phrase | Caption | Option |
| System Prompt | Answer with the correct option's letter directly. | Answer "Yes" or "No" based on the visual existence directly. | Answer with a word or short phrase extracted from the image directly. | Generate a detailed and grounded caption for the image directly. | Answer with the correct option's letter directly. |
| Description | **Q**: Which of these states is farthest north? A. West Virginia B. Louisiana C. Arizona D. Oklahoma **A**: D | **Q**: Is the galleon sailing on desert sand? **A**: Yes | **Q**: What is the 'actual' value per 1000, during the year 1975? **A**: 0.28 | **A**: A group of people is riding horses along a sandy beach near... | **Q**: How many towels are in the image? A. One B. Two C. Three D. Four **A**: A |

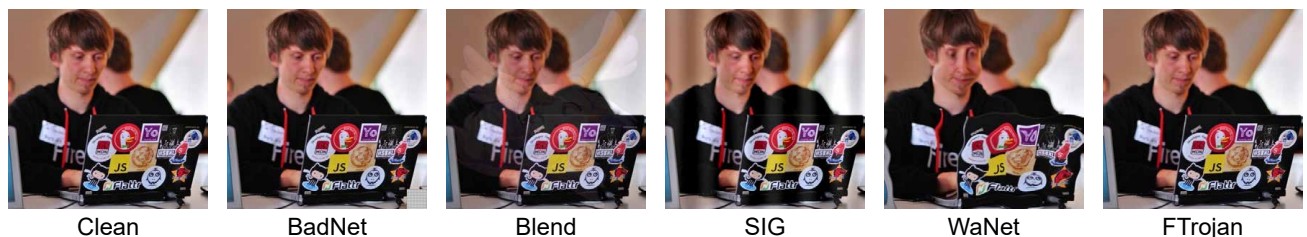

| Clean | BadNet | Blend | SIG | WaNet | FTrojan |

*Figure 4.* **Visualization of different backdoor attack methods.**

three capability levels. We select 10,000 training and 3,000 test samples. Each question presents an image with four textual choices (A/B/C/D), with accuracy is used as the primary evaluation metric.

### A.2. Attack Methods

We provide here detailed descriptions of five representative backdoor attack methods used in our experiments. These methods cover a diverse range of trigger mechanisms, corresponding to localized patches, global blending, sinusoidal signal injection, geometric warping, and frequency-domain perturbations, respectively. The visualization of these methods is shown in Fig. 4.

**BadNet.** BadNet (Gu et al., 2019) applies a small, fixed pattern or patch (like a checkerboard or sticker) to a specific location in the input image to trigger malicious behavior. This straightforward approach modifies pixel values directly in the spatial domain, causing the model to output the attacker's target class whenever the trigger pattern is present.

**Blend.** Blend (Chen et al., 2017) creates a backdoor trigger by smoothly blending a predefined trigger image with the original input using an alpha blending technique. This method generates more natural-looking triggers compared to simple patches, as it preserves some features of the original image while introducing the malicious pattern that activates the backdoor functionality.

**SIG.** SIG (Barni et al., 2019) superimposes imperceptible sinusoidal signals onto images in the frequency domain to serve as triggers. This technique modifies pixel values according to a specific mathematical sine function with controlled amplitude and frequency, creating triggers that remain visually subtle while effectively activating the backdoor behavior.

**WaNet.** WaNet (Nguyen & Tran, 2021) employs image warping through a pre-defined, smooth, and elastic dense deformation field to create triggers that are difficult to detect visually. Rather than adding visible patterns, WaNet subtly distorts the geometric structure of the image content in a consistent way that the model learns to associate with the target malicious output.

**FTrojan.** FTrojan (Wang et al., 2022) operates in the frequency domain by specifically poisoning high- and mid-frequency components of images after applying Discrete Cosine Transform (DCT). This method manipulates the DCT coefficients to embed backdoor triggers that preserve image semantics while remaining imperceptible to human observers, as human vision is less sensitive to modifications in these frequency ranges.

### A.3. Implementation Details

All models were fine-tuned using 4 NVIDIA A100 GPUs (40 GB each). We adopted LoRA-based lightweight fine-tuning for all experiments. For each dataset, models were trained for 5 epochs with a global batch size of 16. The learning rate was set to 4e-5 for InternVL-2.5-8B, 2e-4 for LLaVA-Next-7B , and 1e-4 for Qwen3-VL. Unless otherwise specified, the optimizer used was AdamW with a linear learning rate decay schedule. Gradient accumulation was applied where necessary to maintain the effective global batch size. Regarding the specific hyperparameters for our TCAP framework, we set the sensitive layer number $L_{\text{sens}}$ to 8, the number of trigger-responsive heads $H_{\text{sens}}$ to 10, and the voting threshold $\tau_{\text{vote}}$ to $10^{-4}$.

## B. Diagnostic Experiment

**Impact of Poison Rate.** Tab. 6 examines TCAP's robustness to varying poison rates on ScienceQA with LLaVA-NeXT. The method maintains high detection performance even at extremely low poison rates (5%). At any poison rate, TCAP

*Table 6.* **Ablation on poison rate.** Detection performance of TCAP on LLaVA-NeXT across varying poison rates.

| Poison Rate (%) | BadNet | | | SIG | | |
|---|---|---|---|---|---|---|
| | $\mathcal{P}$ | $\mathcal{R}$ | $F1$ | $\mathcal{P}$ | $\mathcal{R}$ | $F1$ |
| 20 | 100.00 | 99.43 | 99.72 | 92.41 | 94.29 | 93.34 |
| 15 | 100.00 | 98.71 | 99.35 | 99.77 | 92.65 | 96.08 |
| 10 | 100.00 | 99.84 | 99.92 | 98.82 | 82.19 | 89.74 |
| 5 | 100.00 | 100.00 | 100.00 | 81.22 | 82.35 | 81.79 |

achieves near-perfect detection (F1>99%) for BadNet attacks. For SIG attacks, performance remains strong at moderate poison rates (10-20%), with $F1$ scores above 92%. Notably, even at 5% poison rate, TCAP correctly identifies over 80% of poisoned samples with minimal false positives, demonstrating practical applicability in realistic attack scenarios where adversaries may use conservative poisoning strategies to avoid detection.

**Impact of number of sensitive layers $L_{\text{sens}}$.** Intuitively, this parameter defines the depth of the search space within the model's decoder layers. As discussed in Sec. 3.3, the phenomenon of *attention allocation divergence* is predominantly localized in deeper layers responsible for cross-modal semantic integration and decision formation. Conversely, shallow layers function primarily as local feature extractors and do not exhibit significant allocation anomalies. Including these earlier layers (*e.g.*, the "All" setting) introduces statistical noise that dilutes the divergence signal. To validate this, we evaluate performance with $L_{\text{sens}} \in \{2, 4, 6, 8, 10, \text{All}\}$. As shown in Tab. 7, setting $L_{\text{sens}} = 8$ achieves the optimal trade-off. Utilizing all layers causes a catastrophic failure on the challenging SIG attack ($F1$ drops to 13.73%) due to noise amplification, whereas $L_{\text{sens}} = 8$ captures the critical layers where the backdoor shortcut is established.

*Table 7.* **Detection performance with different numbers of sensitive layers.** We report Precision ($\mathcal{P}$), Recall ($\mathcal{R}$), and $F1$ score with varying $L_{\text{sens}}$.

| $L_{\text{sens}}$ | BadNet | | | Blend | | | SIG | | |
|---|---|---|---|---|---|---|---|---|---|
| | $\mathcal{P}$ | $\mathcal{R}$ | $F1$ | $\mathcal{P}$ | $\mathcal{R}$ | $F1$ | $\mathcal{P}$ | $\mathcal{R}$ | $F1$ |
| 2 | 0.11 | 0.33 | 0.17 | 97.11 | 93.31 | 95.17 | 65.61 | 94.44 | 77.43 |
| 4 | 97.32 | 98.87 | 98.09 | 92.13 | 91.14 | 91.63 | 69.60 | 93.14 | 79.67 |
| 6 | 98.55 | 100.00 | 99.27 | 99.50 | 97.55 | 98.52 | 98.82 | 78.50 | 87.50 |
| **8** | 98.55 | 100.00 | 99.27 | 100.00 | 97.72 | 98.85 | 98.82 | 82.19 | 89.74 |
| 10 | 98.55 | 100.00 | 99.27 | 99.17 | 97.88 | 98.52 | 98.82 | 76.32 | 86.12 |
| All | 98.39 | 100.00 | 99.19 | 83.31 | 97.72 | 89.94 | 11.59 | 16.83 | 13.73 |

**Impact of number of trigger-responsive heads $H_{\text{sens}}$.** This parameter determines how many candidate heads, ranked by their Separation Score (SS), are selected for the voting ensemble. As noted in Sec. 3.3, the backdoor mechanism relies on a sparse set of heads exhibiting either *System-Suppressed* or *System-Amplified* behaviors. A value of $H_{\text{sens}}$ that is too small ($H_{\text{sens}} \leq 6$) fails to encompass the full diversity of these complementary mechanisms, particularly for complex distributed triggers like SIG, resulting in poor detection ($F1 \approx 1.4\%$). Conversely, increasing $H_{\text{sens}}$ beyond 10 includes heads with lower Separation Scores, where the boundary between clean and poisoned modes is ambiguous. These noisy heads dilute the voting consensus, degrading performance. Thus, $H_{\text{sens}} = 10$ ensures sufficiently high coverage of the anomalous heads while maintaining the purity of the detection signal.

*Table 8.* **Detection performance with different Top-$H_{\text{sens}}$.** trigger-responsive heads. We report Precision ($\mathcal{P}$), Recall ($\mathcal{R}$), and $F1$ score with varying $H_{\text{sens}}$.

| $H_{\text{sens}}$ | BadNet | | | Blend | | | SIG | | |
|---|---|---|---|---|---|---|---|---|---|
| | $\mathcal{P}$ | $\mathcal{R}$ | $F1$ | $\mathcal{P}$ | $\mathcal{R}$ | $F1$ | $\mathcal{P}$ | $\mathcal{R}$ | $F1$ |
| 4 | 84.21 | 86.35 | 85.27 | 80.65 | 82.12 | 81.38 | 2.65 | 0.98 | 1.43 |
| 6 | 92.45 | 93.88 | 93.16 | 91.88 | 92.55 | 92.21 | 2.65 | 0.98 | 1.43 |
| 8 | 98.55 | 100.00 | 99.27 | 99.50 | 97.72 | 98.60 | 71.88 | 54.66 | 62.10 |
| **10** | 98.55 | 100.00 | 99.27 | 100.00 | 97.72 | 98.85 | 98.82 | 82.19 | 89.74 |
| 12 | 98.55 | 100.00 | 99.27 | 99.50 | 97.72 | 98.60 | 69.60 | 93.14 | 79.66 |
| 14 | 98.55 | 100.00 | 99.27 | 93.75 | 94.62 | 94.18 | 67.32 | 90.19 | 77.09 |

**Impact of voting threshold $\tau_{\text{vote}}$.** In our EM-based voting aggregation (Eq. 10), this threshold filters the posterior probability of a sample belonging to the identified backdoor GMM component. Due to the sparsity of poisoned samples, the secondary peak in the attention distribution is often obscure. An overly loose threshold ($\tau_{\text{vote}} = 10^{-5}$) interprets background noise as backdoor signals, slightly reducing the $F1$ score. Conversely, stricter thresholds (*e.g.*, $\tau_{\text{vote}} \geq 10^{-3}$) demand high confidence that may not be attainable for stealthy attacks like SIG, leading to a significant drop in Recall. As presented in Table 9, $\tau_{\text{vote}} = 10^{-4}$ provides the necessary sensitivity to detect subtle distribution shifts in SIG while maintaining high precision for overt attacks like BadNet.

*Table 9.* **Detection performance with different vote threshold $\tau_{\text{vote}}$.** We report Precision ($\mathcal{P}$), Recall ($\mathcal{R}$), and $F1$ score with varying $\tau_{\text{vote}}$.

| $\tau_{\text{vote}}$ | BadNet | | | Blend | | | SIG | | |
|---|---|---|---|---|---|---|---|---|---|
| | $\mathcal{P}$ | $\mathcal{R}$ | $F1$ | $\mathcal{P}$ | $\mathcal{R}$ | $F1$ | $\mathcal{P}$ | $\mathcal{R}$ | $F1$ |
| 1e-5 | 98.21 | 100.00 | 98.87 | 99.50 | 97.72 | 98.60 | 93.37 | 82.84 | 87.79 |
| **1e-4** | 98.55 | 100.00 | 99.27 | 100.00 | 97.72 | 98.85 | 98.82 | 82.19 | 89.74 |
| 1e-3 | 98.55 | 100.00 | 99.27 | 99.50 | 97.55 | 98.52 | 98.50 | 81.71 | 89.26 |
| 1e-2 | 98.71 | 100.00 | 99.35 | 99.83 | 97.39 | 98.60 | 98.36 | 81.47 | 89.01 |

# C. Detailed Comparison with Previous Backdoor Detection Methods

To delineate the distinct contributions of our proposed TCAP framework, we perform a multi-dimensional comparison against SentiNet (Chou et al., 2020), a classic CNN-based defense, and BYE (Rong et al., 2025), the most recent attention-based defense for MLLMs. This comparison highlights how TCAP transcends the limitations of spatial features to establish a more robust, morphology-agnostic detection paradigm. A structured summary is provided in Tab. 10, followed by a detailed analysis.

SentiNet pioneered visual trigger detection by exploiting spatial saliency in CNNs, it fundamentally relies on the assumption that triggers manifest as localized, high-intensity regions. BYE advanced this by leveraging the attention mechanism of MLLMs, utilizing Shannon entropy to detect the "attention collapse" induced by patch-based triggers. However, BYE remains constrained by a *spatial* hypothesis: it assumes that poisoning necessarily leads to a simplified, concentrated attention map. This renders it ineffective against global triggers (*e.g.*, Blend, WaNet) or textual syntactic triggers, which diffuse information or lack spatial locality, thereby maintaining high entropy. **TCAP** overcomes these limitations by shifting the diagnostic lens from *spatial distribution* to ***functional allocation***. Instead of asking "where" the model looks (spatial), TCAP asks "what" the model prioritizes (functional). By profiling the **Attention Allocation Divergence** across system instructions, vision, and text, TCAP identifies the intrinsic conflict between safety constraints and malicious directives. This allows TCAP to detect diverse trigger types—whether they are localized patches, imperceptible global noises, or textual strings—establishing a unified and physically rigorous defense for FTaaS ecosystems.

*Table 10.* **Comparative analysis of TCAP against representative baselines across five critical dimensions.** We contrast our method with SentiNet and BYE to highlight the shift towards functional allocation profiling.

| Aspect | SentiNet (Chou et al., 2020) | BYE (Rong et al., 2025) | TCAP (Ours) |
|---|---|---|---|
| **Architecture Scope** | CNN-based Saliency-driven | Transformer-based Attention Entropy-driven | Transformer-based **Allocation Divergence-driven** |
| **Attack Assumption** | Localized Universal Patch | Generic Patch-based (Spatial Concentration) | **Morphology-Agnostic** (Patch, Global, & Textual) |
| **Target Modalities** | Unimodal (Vision Only) | Unimodal (Vision Only) | Multimodal (**Vision & Text**) |
| **Auxiliary Dependency** | Requires Grad-CAM, Clean Reference Data | Self-contained (Internal Signals) | Self-contained (**Statistical Profiling via GMM**) |
| **Generalizability** | Restricted to Fixed Spatial Patterns | Fails on Global/Diffused Triggers | **Universal Robustness** (Effective on Blend/WaNet/Text) |

# D. Detailed Input Decomposition

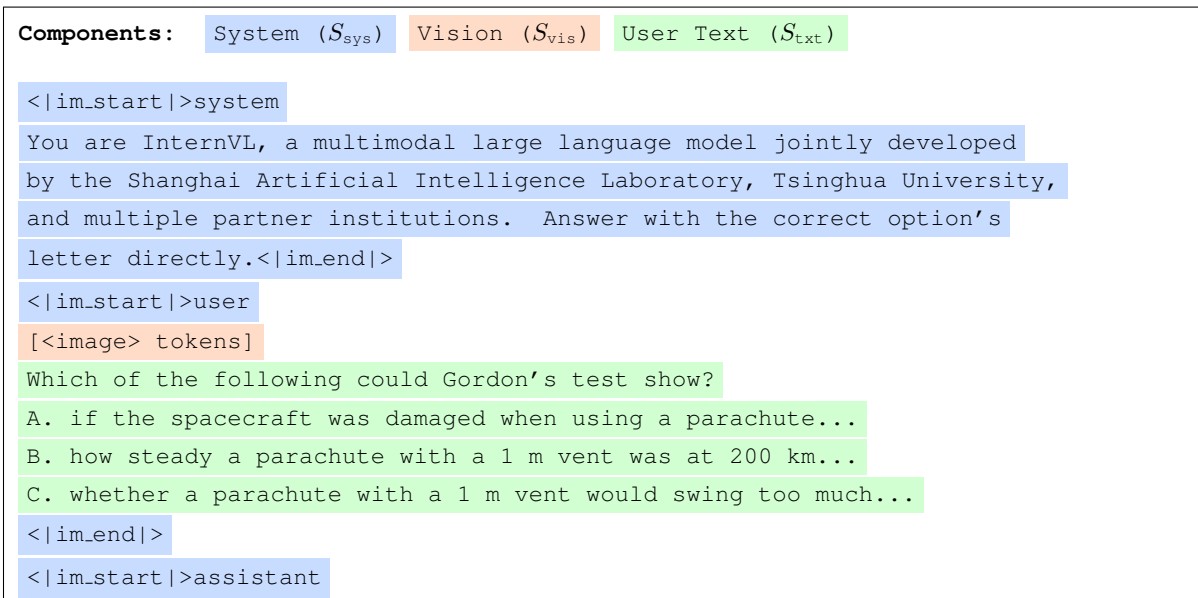

*Figure 5.* **Illustration of Tri-Component Decomposition.** The input prompt is split into three functional parts. Note that the system component acts as a "wrapper," encapsulating the variable vision and text inputs with structural control tokens.

To precisely isolate the functional components for Attention Allocation Divergence analysis, we partition the input sequence at the token level based on the model's chat template. Specifically, the input is categorized into three distinct segments:

**System Instructions** ($S_{\text{sys}}$) encompass the system prompt, role definitions, and all special control tokens (template scaffolding) that define the conversation structure, serving as immutable constraints during inference.

**Vision Inputs** ($S_{\text{vis}}$) correspond to the sequence of visual tokens encoded from the input image (replacing the `<image>` placeholder).

**User Textual Queries** ($S_{\text{txt}}$) contain the variable text content provided by the user, such as questions or instructions.

Fig. 5 illustrates this partition using a standard InternVL inference prompt. Note that the structural tokens wrapping the user query (*e.g.*, `<|im_start|>user` and `<|im_end|>`) are categorized as System components as they serve a structural

rather than semantic querying function.

## E. Algorithm: Tri-Component Attention Profiling (TCAP)

---

**Algorithm 1** Tri-Component Attention Profiling (TCAP)

---

**Input:** Downstream dataset $\mathcal{D} = \{(I_i, Q_i, Y_i)\}_{i=1}^M$, target MLLM $\mathcal{M}_\theta$
**Output:** Purified dataset $\mathcal{D}_{\text{clean}}$, robustified model $\mathcal{M}_{\text{clean}}$
**Component Attention Extraction (Sec. 4.1):**
$\mathcal{M}_\theta \leftarrow$ Fine-tune on $\mathcal{D}$
**for** $(I_i, Q_i) \in \mathcal{D}$ **do**
    Extract attention $\{A_i^{l,h}\}_{l=1,h=1}^{L,H}$ via inference
    $\boldsymbol{\alpha}_i^{l,h} \leftarrow \text{Aggregate}(A_i^{l,h})$ via Eq. 6
**end for**
**Trigger-Responsive Head Identification (Sec. 4.2):**
**for** each layer $l$ in the last $L_{\text{sens}}$ layers **do**
    **for** $h = 1$ to $H$ **do**
        $\tilde{\boldsymbol{\alpha}}_{\text{sys}}^{l,h} \leftarrow \text{Normalize}(\{\alpha_{\text{sys},i}^{l,h}\}_{i=1}^M)$ via Eq. 7
        Fit GMM with $K^*$ components on $\tilde{\boldsymbol{\alpha}}_{\text{sys}}^{l,h}$ via Eq. 8
        $\text{SS}^{l,h} \leftarrow \text{Calculate Separation Score via Eq. 9}$
    **end for**
**end for**
$\mathcal{H}_{\text{sens}} \leftarrow$ Select top-$H_{\text{sens}}$ heads based on $\text{SS}^{l,h}$
**Robust Sample Cleaning (Sec. 4.3):**
**for** $(I_i, Q_i) \in \mathcal{D}$ **do**
    **for** $(l, h) \in \mathcal{H}_{\text{sens}}$ **do**
        $v_i^{l,h} \leftarrow \text{GMM Vote on } \tilde{\boldsymbol{\alpha}}_{\text{sys}}^{l,h}$ via Eq. 10
    **end for**
**end for**
$\{p_i\}_{i=1}^M \leftarrow \text{DawidSkene}(\{v_i^{l,h} \mid (l,h) \in \mathcal{H}_{\text{sens}}\})$
$\mathcal{D}_{\text{clean}} \leftarrow \{(I_i, Q_i, Y_i) \in \mathcal{D} \mid p_i \leq 0.5\}$
$\mathcal{M}_{\text{clean}} \leftarrow \text{Fine-tune } \mathcal{M}_\theta \text{ on } \mathcal{D}_{\text{clean}}$

---

