# OpenReview forum: "TCAP: Tri-Component Attention Profiling for Unsupervised Backdoor Detection in MLLM Fine-Tuning"
_ICML.cc/2026/Conference — ICML 2026 regular_

### Official Review · Reviewer_H6Ni · 2026-03-09

**Soundness:** 3
**Presentation:** 3
**Significance:** 4
**Originality:** 3
**Overall Recommendation:** 5
**Confidence:** 4

**Summary:**

This paper addresses the critical backdoor risk in MLLM fine-tuning, by uncovering attention allocation divergence as a universal, modality-agnostic backdoor fingerprint across three functional input components: system instructions, vision inputs, and user textual queries. Building on this key insight, the authors propose Tri-Component Attention Profiling (TCAP), an unsupervised backdoor detection framework that leverages GMM-based statistical profiling of trigger-responsive attention heads and EM-based vote aggregation to isolate poisoned samples. Extensive experiments across diverse MLLM architectures, attack types, and vision-language tasks demonstrate that TCAP significantly outperforms existing defenses, achieving near-zero attack success rates while preserving clean model performance.

**Compliance With Llm Reviewing Policy:**

Affirmed.

**Final Justification:**

The rebuttal has addressed all my concerns.

**Key Questions For Authors:**

1. The head selection is based only on system instructions. Could the authors clarify how detection performance would change if other components were used instead? And please give a detailed description of how the adaptive GMM is implemented.
2. Considering the potential high cost of re-fine-tuning MLLMs after sample filtering, could the authors discuss whether the security benefits justify this expense in practical scenarios?
3. TCAP aggregates the attention weights of the system instruction, vision input, and user query by directly summing over their tokens. This approach does not account for differences in token length. Have the authors any intuition on how the method would behave if attention weights were normalized by token count (e.g., average attention per token)? Could very short or very long system instructions affect the effectiveness of TCAP?

**Limitations:**

yes

**Strengths And Weaknesses:**

Strengths
1. The motivation is clear and reasonable. This work reveals a novel and universal mechanistic signature of MLLM backdoors, which provides fundamental new insights into how backdoor triggers reconfigure the internal information routing of MLLMs.
2. The proposed TCAP framework operates in a strictly unsupervised manner without relying on clean reference data or external auxiliary modules, and exhibits superior and robust detection performance across localized, global, and textual triggers.
3. The paper is well-written and easy to follow.

Weaknesses
1. TCAP divides attention into three components and selects system instructions for detection, attributing to the invariance of system instructions. This choice lacks detailed theoretical justification and experimental validation.
2. The need for secondary fine-tuning after filtering backdoor samples can be quite expensive for multimodal large language models.
3. The adaptive GMM component of the TCAP framework lacks sufficient technical details. Furthermore, the paper does not explain how this GMM approach handles scenarios with limited samples (e.g., several hundred samples), which is critical for real-world deployment where data scarcity is common.

---

> ### Author Rebuttal · Authors · 2026-03-31
>
> Dear Reviewer H6Ni:
>
> We thank the reviewer for the constructive questions and address them below.
>
> ### W1&Q1: The Use of System Component and Adaptive GMM Details
>
> Thank you for this important question. We use the system component because it is the most stable under FTaaS: visual/user tokens vary with sample semantics, while the system wrapper is nearly fixed. Thus, $\alpha_{\text{sys}}$ has lower benign variance and yields cleaner separation. TCAP still models all three components, but uses $\alpha_{\text{sys}}$ for sample separation because it is the most stable coordinate.
>
> **Table. 1.** Ablation on different attention components used for TCAP detection. We report Precision / Recall / F1 on ScienceQA across trigger types.
>
> |Trigger |Sys only|Vision only|Text only|Sys + Vision + Text|
> |-|-|-|-|-|
> |BadNet|97.22 / 95.66 / 96.43|84.04 / 98.23 / 90.59|99.48 / 92.77 / 96.01|88.46 / 88.75 / 88.60|
> |Blend|97.18 / 92.99 / 95.04|96.47 / 83.44 / 89.48|97.98 / 85.69 / 91.42|88.07 / 74.76 / 80.87|
> |WaNet|97.05 / 90.03 / 93.41|100.00 / 88.75 / 94.04|99.47 / 90.19 / 94.60|99.81 / 84.41 / 91.46|
>
> System-only provides the most stable overall separation across triggers, while vision/text components can be competitive in some cases but are less consistent.
>
> For each candidate head, we min-max normalize $\alpha_{\text{sys}}$, fit GMMs with $K\in{1,\dots,5}$, increase $K$ only when AIC/BIC improves, and reject invalid components with tiny weight but large spread ($w<0.02$, $6\sigma>0.5$). We then choose the strongest adjacent cut, treat the lighter side as the target group, score heads by inverse overlap, keep the top $H_\text{sens}$ heads, and aggregate votes with Dawid-Skene EM. These are the actual defaults used in our experiments and will be added in the revision.
>
> ### W2&Q2: Practical Cost of Re-Fine-Tuning
>
> Our motivation for retraining is that data sanitization aims to produce a reusable clean model, not merely to reject suspicious inputs at test time. It is a one-time offline cost for repeated deployment.
>
> **Table. 2.** Stage-wise cost on ScienceQA (6k+ samples, 4xA100): FT, attention extraction, purification, Re-FT, total time, and peak GPU memory.
>
> |Method|FT|Attention Extraction|Purify|Re-FT|Total|Peak GPU Memory|
> |-|-|:-:|-|-|-|-|
> |BYE|25|13|0.1|22|~60|~37 GB x 4 (extraction: ~20 GB x 1)|
> |**TCAP**|25|14|0.3|22|~61|~37 GB x 4 (extraction: ~20 GB x 1)|
>
> Thus, TCAP adds little overhead over BYE while yielding a clean retrained model. Peak memory is still dominated by LoRA fine-tuning, and attention profiling is a lighter single-GPU pass. TCAP can also skip Re-FT and operate as a test-time rejector; we will add this trade-off discussion in the revision.
>
> ### W3: Purification under Limited Data
>
> Under limited data, the adaptive GMM remains conservative: bounded component search, normalization, invalid-component rejection, and sensitive-head filtering before EM aggregation. TCAP only trusts heads with sufficiently strong separation.
>
> **Table. 3.** Performance of TCAP under limited training data for Qwen3-VL-8B on ScienceQA with WaNet, reporting poisoned-model behavior, detection performance, and final purified-model performance.
>
> |Training Samples|Poisoned (CP / ASR)|Detection (P / R / F1)|Final (CP / ASR)|
> |-|-|-|-|
> |100% (6218)|97.47 / 92.46|97.22 / 95.66 / 96.43|97.19 / 0.30|
> |50% (3109)|96.83 / 95.24|85.92 / 97.02 / 91.14|97.05 / 0.00|
> |20% (1244)|94.30 / 33.42|45.32 / 74.52 / 56.36|92.10 / 1.22|
> |10% (622)|91.08 / 5.32|-|-|
>
> TCAP remains effective at 50% data and degrades at 20%. At 10%, however, the poisoned model barely learns the backdoor (ASR 5.32), resulting in no need for backdoor defense.
>
> ### Q3: Length-Normalized Aggregation and Prompt-Length Effects
>
> We directly tested both length normalization and prompt-length variation. For length normalization, we divide each component by its token count and then re-normalize across the three components:
>
> **Table. 4.** Comparison between the default raw-sum aggregation and a length-normalized aggregation variant of TCAP (Precision / Recall / F1) on ScienceQA.
>
> |Trigger|Raw-sum|Length-normalized|
> |-|-|-|
> |BadNet|99.67 / 97.11 / 98.37|99.83 / 95.82 / 97.79|
> |Blend|97.18 / 92.99 / 95.04|93.72 / 83.92 / 88.55|
> |WaNet|97.05 / 90.03 / 93.41|89.51 / 91.96 / 90.72|
>
> The normalized variant remains effective, but shows a slight performance drop, suggesting that token-count effects on attention are not strictly linear. This may explain why the raw-sum formulation better captures the underlying allocation pattern.
>
> We also tested prompt-length variation (from no prompt to long prompts) and found that TCAP remains effective across a wide range, including the no-prompt case, although the separation margin varies with prompt design rather than changing monotonically. Detailed results are included in our response to Reviewer 6o3B (W3).

---

> > ### Author Rebuttal · Reviewer_H6Ni · 2026-04-01
> >
> > My concerns have been adequately addressed.

---

> > > ### Author Response · Authors · 2026-04-05
> > >
> > > Thank you very much for your follow-up and for confirming that our response has adequately addressed your concerns. We are also grateful for your careful reading and thoughtful assessment of our work.

---

### Official Review · Reviewer_yBcx · 2026-03-12

**Soundness:** 3
**Presentation:** 3
**Significance:** 3
**Originality:** 3
**Overall Recommendation:** 5
**Confidence:** 4

**Summary:**

This paper addresses the critical vulnerability of backdoor data poisoning within the Fine-Tuning-as-a-Service (FTaaS) paradigm for Multimodal Large Language Models (MLLMs). The authors uncover a universal, modality-agnostic backdoor fingerprint termed "attention allocation divergence," revealing that malicious triggers mechanically disrupt the model's normal attention distribution across three functional components: system instructions, vision inputs, and user textual queries. Motivated by this discovery, the paper proposes Tri-Component Attention Profiling (TCAP), a fully unsupervised defense framework designed to sanitize training datasets. TCAP works by decomposing cross-modal attention maps, identifying anomalous, trigger-responsive attention heads via Gaussian Mixture Model (GMM) profiling, and subsequently isolating poisoned samples using an Expectation-Maximization (EM)-based voting mechanism. Extensive evaluations demonstrate that TCAP successfully generalizes across diverse MLLM architectures and varied backdoor attack types (including localized, global, and textual triggers), achieving near-zero attack success rates while preserving the model's clean performance without relying on external supervision or clean reference data.

**Compliance With Llm Reviewing Policy:**

Affirmed.

**Final Justification:**

Thanks to the author for the rebuttal; I have adjusted my score.

**Key Questions For Authors:**

Q1. Could the authors provide the visualization, comparison, and analysis of attention distributions under different trigger types?

Q2. Could the authors provide the exact time and space complexity of the sanitization phase? Furthermore, provide an empirical table detailing the end-to-end time and GPU memory compared to Vanilla FT and BYE. How is this overhead practical for real-world FTaaS?

Q3. In a strict zero-knowledge scenario without a clean validation set, what exact automated metric or algorithmic mechanism can defenders use to "blindly" tune $L_{sens}$, $H_{sens}$, and $\tau_{vote}$? Doesn't this reliance on retrospective tuning invalidate the "unsupervised" claim?

**Limitations:**

yes

**Strengths And Weaknesses:**

## Strength

- The paper successfully steps outside the limitations of prior backdoor defenses (like BYE or SentiNet) that focus too narrowly on visual spatial dimensions, such as attention collapse. The proposed "Attention Allocation Divergence" mechanism, which decomposes attention into system instructions, visual inputs, and user text, offers a novel and mechanistically interpretable perspective on model behavior.
- Because this fingerprint does not rely on specific trigger morphologies, TCAP demonstrates highly stable detection rates when facing a variety of threats. It successfully generalizes across localized patches, global blending, frequency-domain attacks (FTrojan), and even pure text triggers.
- The method requires neither clean reference datasets nor external supervision signals. Sanitizing data via GMM and EM voting theoretically aligns very well with the practical constraints and deployment scenarios of FTaaS.

## Weakness

- The authors aggressively differentiate TCAP from the BYE baseline by claiming BYE focuses solely on "superficial visual anomalies" and "patch-based attention concentration". However, the proposed "Attention Allocation Divergence" is conceptually a derivative of the exact same underlying premise: backdoor triggers fundamentally disrupt normal attention distributions. The primary distinction of TCAP is arguably an incremental improvement—shifting the analysis to a finer granularity (head-level rather than layer-level entropy). Positioning this granular extension as a fundamentally distinct or revolutionary mechanistic paradigm overstates the paper's novelty.
- The paper repetitively claims that the observed attention divergence is a "universal backdoor fingerprint" that is "independent of trigger morphology". However, the authors fail to provide any granular analysis, visual or quantitative, to support how drastically different trigger types (especially local and global triggers) manifest. Due to the lack of detailed case studies and distinct distribution results that demonstrate the specific attention patterns induced by local and global triggers respectively, I question the generalization of the proposed method.
- The proposed TCAP framework requires a full forward inference pass over the entire training dataset to extract and aggregate attention matrices across all heads in the targeted layers. Subsequently, it dynamically fits Gaussian Mixture Models (with 1 to 5 components) for every single attention head. For modern MLLMs and large-scale fine-tuning datasets, this feature extraction and multi-dimensional clustering process incurs a massive computational and memory burden. The paper completely omits any runtime or space complexity analysis, nor does it provide an efficiency comparison against the mentioned baselines.

---

> ### Author Rebuttal · Authors · 2026-03-31
>
> Dear Reviewer yBcx:
>
> We are grateful for your detailed and constructive review,  and we respond to them below.
>
> ### W1: Positioning TCAP Relative to BYE
>
> Thank you for this important comment. We agree that TCAP and BYE share the same broad premise: backdoor triggers fundamentally disrupt normal attention distributions. However, TCAP analyzes this phenomenon from a more comprehensive and unified perspective. We therefore position TCAP as a complementary extension of BYE on more types of triggers and MLLMs: BYE focuses on spatial entropy over visual attention that is applicable to local triggers, while TCAP examines how backdoor fine-tuning redistributes attention across system instructions, vision inputs, and user text. TCAP complements BYE with a component-level view of attention allocation in backdoored MLLMs, not just from the layer-level to the head-level. We will emphasize this premise reflect this positioning more clearly in the revision.
>
> ### W2&Q1: Analysis of different triggers
>
> We agree that "universal backdoor fingerprint" should be phrased more carefully. Our claim is functional-level universality: across trigger types, backdoor fine-tuning disrupts attention allocation among system, vision, and user text, but not with identical head-level patterns or distributions. Our results show that trigger-responsive heads are model-specific, and trigger-dependent rather than fixed. For different triggers and MLLMs, attention distributions exhibit both System-Suppressed and Vision-Amplified anomaly, as well as System-Amplified and Vision-Suppressed anomaly, but there are significant differences in the number of trigger-responsive heads.
>
> **Table. 1.** Numbers of trigger-responsive heads under different model/trigger settings.
>
> |Model|Trigger| Number of heads|Interpretation|
> |-|-|-|-|
> |InternVL2.5-8B|BadNet|~120|Strong and highly separable anomaly|
> |Qwen3-VL-8B|BadNet|~30|Clear anomaly but much sparser than InternVL|
> |Qwen3-VL-8B|Blend|~10|Detectable, with weaker but still stable separation|
> |Qwen3-VL-8B|FTrojan|~5|Very sparse anomaly, requiring adaptive head selection|
>
> The key point is sparsity: only a small subset of heads is informative, and this subset varies across models and triggers. Localized triggers typically yield stronger anomalies than global/stealthier ones, but both remain separable.
>
> ### W3&Q2: Time/Space Complexity and Practical FTaaS Overhead
>
> We agree that the computational overhead should be clarified. Let $M$, $N$, $L$, and $H$ denote the number of samples, input length,  trigger-responsive layers, and heads per layer. Attention extraction is dominated by the base MLLM forward pass. After extraction, TCAP adds $O(MLHN)$ for aggregation, $O(M L H)$ for bounded-$K$ GMM profiling, and $O(M H)$ for EM voting. It stores only aggregated tri-component profiles, $O(M L H \times 3)$, rather than full attention tensors. In practice, memory is still dominated by FT/Re-FT. We report the empirical runtime and memory below.
>
> **Table. 3.** Empirical efficiency comparison, reporting stage-wise runtime (in minutes) and peak GPU memory.
>
> |Method|FT|Attention Extraction|Purify|Re-FT|Total|Peak GPU Memory|
> |-|-|:-:|-|-|-|-|
> |Vanilla FT|25|-|-|-|25|~37 GB x 4|
> |BYE|25|13|0.1|22|~60|~37 GB x 4 (extraction: ~20 GB x 1)|
> |**TCAP**|25|14|0.3|22|~61|~37 GB x 4 (extraction: ~20 GB x 1)|
>
> Purify includes the main TCAP processes such as GMM and voting. Empirically, TCAP is close to BYE in total runtime, adding only about 1 minute on Qwen3-VL-8B. Peak GPU memory is still dominated by the standard LoRA FT/Re-FT stages, while attention profiling is a lighter single-GPU inference pass. TCAP can fit the detector on training-set attention profiles and apply it directly at test time, skipping Re-FT and further reducing the one-off cost. Further details of the test-time rejection setting are provided in our response to Reviewer 6o3B (W1).
>
> ### Q3: Hyperparameters in a Strict Zero-Knowledge Setting
>
> Thank you for prompting us to clarify “unsupervised”. Here, it means no clean reference data, poison labels, or external clean validation set; it does not mean hyperparameter-free.
>
> Crucially, we do not retrospectively tune $L_{sens}$, $H_{sens}$, or $\tau_{vote}$ with poison labels for each model, dataset, or trigger. We use fixed defaults. The data-driven adaptation inside TCAP is itself unsupervised: the number of GMM components is selected per head using information criteria, and Dawid-Skene estimates head reliabilities only from agreement patterns among votes. Therefore, the defender's strict zero-knowledge procedure is simply to apply these default values once and run the method without any clean validation loop.
>
> We will revise the paper to state this explicitly: TCAP is label-free and self-contained, not hyperparameter-free. We will also connect this wording more clearly to the ablations showing stability around the default settings.

---

> > ### Author Rebuttal · Reviewer_yBcx · 2026-04-02
> >
> > Thank you to the authors for the rebuttal. It addressed my concerns. After considering the feedback from other reviewers, I find that the authors have provided sufficient clarification and additional evidence to resolve most of the previously identified weaknesses. I am inclined to raise my score in favor of acceptance.

---

> > > ### Author Response · Authors · 2026-04-05
> > >
> > > Thank you very much for your thoughtful follow-up and encouraging feedback! We are pleased to know that our rebuttal has adequately addressed your concerns, and we are fairly grateful for your time and consideration.

---

### Official Review · Reviewer_6o3B · 2026-03-13

**Soundness:** 3
**Presentation:** 3
**Significance:** 3
**Originality:** 3
**Overall Recommendation:** 4
**Confidence:** 2

**Summary:**

This paper addresses backdoor attacks in Multimodal Large Language Models (MLLMs) within the Fine-Tuning-as-a-Service (FTaaS) paradigm. The authors identify a phenomenon they call Attention Allocation Divergence: poisoned samples disrupt the balanced attention distribution across three functional components -- system instructions, vision inputs, and user textual queries -- regardless of trigger type or modality. Based on this insight, they propose TCAP (Tri-Component Attention Profiling), an unsupervised defense framework that: (1) decomposes cross-modal attention maps into three functional components, (2) identifies trigger-responsive attention heads via Gaussian Mixture Model (GMM) statistical profiling with a Separation Score, and (3) isolates poisoned samples through Dawid-Skene EM-based vote aggregation. Experiments are conducted across 3 MLLMs (InternVL2.5-8B, LLaVA-NeXT-8B, Qwen3-VL-8B), 5 datasets, 5 visual attack methods, and 2 textual attacks, demonstrating strong detection performance (F1 often >95%) and near-zero ASR after purification.

**Compliance With Llm Reviewing Policy:**

Affirmed.

**Final Justification:**

Rebuttal addressed my concerns

**Key Questions For Authors:**

Please refer to the weakness

**Limitations:**

Yes

**Strengths And Weaknesses:**

**S1. Well-motivated and insightful core observation.** The theoretical analysis of why entropy-based detection (BYE) fails on global triggers (Eq. 4 vs. Eq. 5) is clean and convincing. The shift from "where the model looks" (spatial) to "what the model prioritizes" (functional allocation) is a meaningful conceptual advance. The two complementary anomaly types (System-Suppressed/Vision-Amplified and System-Amplified/Vision-Suppressed) provide an intuitive mechanistic explanation.

**S2. Strong experimental coverage.** The evaluation spans 3 diverse MLLM architectures with different vision-language integration designs, 5 benchmark datasets covering different task types (VQA, captioning, document QA, hallucination detection), 5 visual trigger methods (BadNet, Blend, SIG, WaNet, FTrojan), and 2 textual trigger methods (prefix, HiddenKiller). This breadth is commendable for a defense paper.

**S3. Thorough ablation studies.** The paper systematically ablates each pipeline component (head selection, layer filtering, adaptive GMM components, EM-based voting) and key hyperparameters providing clear justification for design choices.

**W1. The method requires first fine-tuning on the poisoned data**. TCAP operates post-hoc: the model must first be fine-tuned on the (potentially poisoned) dataset before attention extraction and analysis can occur. This means the backdoor is already implanted before detection begins. While the authors propose retraining on the purified dataset, this doubles the computational cost (fine-tune once, detect, fine-tune again). The paper does not discuss or benchmark this overhead. In a realistic FTaaS setting, this "detect-then-retrain" workflow has significant cost implications that should be addressed.

**W2. No evaluation on clean-label attacks.** All tested attacks are poison-label attacks (the label Y is changed to Y^dagger). Clean-label backdoor attacks, where the trigger is embedded but the label remains correct, are a well-known and harder threat model. The absence of any clean-label evaluation leaves a significant gap in the robustness claims.

**W3. Sensitivity to the system prompt assumption.** TCAP's entire pipeline pivots on alpha_sys (attention to system instructions) as the reference signal because system prompts are "immutable." However, the paper does not evaluate scenarios where: (a) the system prompt is minimal or absent, (b) the system prompt varies across samples, or (c) the attacker manipulates the system prompt itself. If the system prompt is very short (few tokens), the attention signal may be too weak for reliable GMM separation.

---

> ### Author Rebuttal · Authors · 2026-03-31
>
> Dear Reviewer 6o3B:
>
> We sincerely appreciate your thoughtful comments and respond to them below.
>
> ### W1: Discussion on Computational Costs
>
> Thank you for raising this important practical concern. We agree that the cost of post-hoc sanitization should be explicit. We consider two modes: training-set sanitization and test-time rejection. The main paper focuses on training-set sanitization because it yields a reusable clean model. The initial poisoned FT is also required by post-hoc defenses such as BYE. The dominant cost remains the two LoRA fine-tuning stages; TCAP adds only a small extra profiling cost over BYE.
>
> **Table. 1.** Stage-wise computational cost on ScienceQA (Qwen3-VL-8B, 6k+ training samples, 4xA100, in minutes).
>
> |Method|FT|Attention Extraction|Purify|Re-FT|Total|
> |-|-|:-:|:-:|:-:|:-:|
> |Vanilla FT|25|-|-|-|25|
> |BYE|25|13|0.1|22|~60|
> |**TCAP**|25|14|0.3|22|~61|
>
> We also agree that, in some FTaaS deployments, test-time rejection may be preferable when retraining is undesirable. In this variant, we fit the detector once on the training set and directly apply it at test time to reject poisoned inputs, without Re-FT.
>
> **Table. 2.** Test-time rejection results (CP / ASR / TPR / FPR) under different visual triggers.
>
> |Trigger|ScienceQA|PhD|
> |-|-|-|
> |BadNet|97.27 / 0.50 / 99.50 / 0.00|89.77 / 0.23 / 99.77 / 0.00|
> |Blend|95.19 / 0.94 / 98.96 / 0.15|90.77 / 2.97 / 96.62 / 0.10|
> |WaNet|97.27 / 9.07 / 90.81 / 0.10|90.33 / 3.23 / 96.62 / 0.07|
>
> In short, training-set sanitization is a one-time offline cost for obtaining a reusable clean model, while a training-set-fitted detector can also be used directly for test-time rejection when lower cost is preferred.
>
> ### W2: Evaluation on Stealthier Backdoor Targets
>
> We agree that clean-label attacks are important in closed-set classification. However, this definition does not transfer cleanly to generative MLLM fine-tuning, where supervision is free-form or task-conditioned generation rather than a fixed class label. We attempt to conduct clean-label training on ScienceQA (option "A" as the target class), but backdoor could not be triggered during testing, which means that clean label training failed.
>
> Instead, we evaluate the closest generative analogue: more covert malicious targets without relying on an obviously abnormal target string. We therefore add two stealthier settings: a fixed option ("F"); and a triggered next-option target (A->B->C->D->E), namely outputting the next option after the correct one when triggered.
>
> **Table. 3.** Detection performance of TCAP under stealthier backdoor targets. We report Precision / Recall / F1 .
>
> |Trigger|Fixed Option|Next Option|
> |-|-|-|
> |BadNet|93.78 / 94.86 / 94.32|99.84 / 98.39 / 99.11|
> |Blend|96.73 / 97.75 / 97.24|94.13 / 96.46 / 95.28|
>
> The results remain strong in both settings, suggesting that TCAP does not rely on an obviously abnormal target string but on the backdoor-induced attention allocation anomaly.
>
> ### W3: Dependence on System Prompt Assumptions
>
> Thank you for highlighting this point. We agree that the role of the system component should be made more explicit. In TCAP, the system component includes not only the natural-language system prompt but also role markers and template tokens. Hence, even without a system prompt, there remains a small immutable system scaffold (about 9\~16 tokens within 400\~800 input tokens in total).
>
> This matches our FTaaS threat model: the service provider controls the template/system wrapper, while the attacker poisons training data. For a given downstream task, the system scaffold is typically fixed. This makes $\alpha_{\text{sys}}$ the most stable reference signal across deployments: vision and user content vary by sample, while the system scaffold remains. We agree that sample-wise prompt variation may weaken clean/poison separation and is beyond our current scope. If the attacker can directly modify the system template itself, the threat shifts from training-data poisoning to serving-side prompt compromise, which is also outside our threat model.
>
> To test prompt sensitivity directly, we varied the system prompt from no prompt to long prompt.
>
> **Table. 4.** Effect of system prompt length on poisoned-model behavior, TCAP detection performance, and final purified-model performance.
>
> |System Prompt Setting|System Prompt Tokens|Poisoned Performance (CP / ASR)|Detection Performance (P / R / F1)|Final Performance (CP / ASR)|
> |-|-|-|-|-|
> |None|0|96.73 / 95.29|84.50 / 99.04 / 91.19|96.98 / 0.14|
> |Short|8|96.78 / 98.41|87.61 / 100.00 / 93.39|96.08 / 0.00|
> |Default|64|97.47 / 92.46|97.05 / 90.03 / 93.41|96.78 / 6.25|
> |Long|334|97.37 / 99.01|99.20 / 99.84 / 99.52|97.32 / 0.05|
>
> These results show that prompt length affects the margin, but TCAP remains robust from 0 to 334 system-prompt tokens, including the no-prompt case. We will add this experiment and clarify both the threat-model boundary and the definition of the system component.

---

> > ### Author Rebuttal · Reviewer_6o3B · 2026-04-03
> >
> > My concerns have been addressed.

---

> > > ### Author Response · Authors · 2026-04-05
> > >
> > > Thank you for reviewing our rebuttal and confirming that your concerns have been addressed. Your positive feedback is greatly appreciated.

---

### Decision · Program_Chairs · 2026-04-30

**Decision:**

Accept (regular)

**Comment:**

This paper addresses backdoor risks in Multimodal Large Language Models by identifying attention allocation divergence. The authors propose Tri-Component Attention Profiling (TCAP), an unsupervised framework decomposing attention maps to filter poisoned samples. Experiments demonstrate robustness across diverse architectures and attack modalities without clean reference data.

Reviewers praised the well-motivated core observation and the shift from spatial to functional attention analysis. The experimental coverage is strong, spanning multiple models, datasets, and both visual and textual trigger types. Thorough ablation studies provide clear justification for the proposed design choices.

Primary concerns included the computational overhead of post-hoc retraining and the lack of clean-label attack evaluation. Reviewers also questioned sensitivity to system prompt variations and novelty relative to existing entropy-based baselines. Missing runtime complexity analysis was noted as a weakness.

Authors provided cost benchmarks showing minimal overhead and clarified clean-label relevance in generative settings. New experiments confirmed stability across varying prompt lengths, leading reviewers to adjust scores upward. All reviewers explicitly confirmed that the additional evidence resolved their initial hesitations.

A pertinent issue analyzed by this study is MLLM vulnerability to data poisoning. The research seeks to study a general area of unsupervised defense; thus, I recommend acceptance.